# The Effects of Tannins in Monogastric Animals with Special Reference to Alternative Feed Ingredients

**DOI:** 10.3390/molecules25204680

**Published:** 2020-10-14

**Authors:** Zahra Mohammed Hassan, Tlou Grace Manyelo, Letlhogonolo Selaledi, Monnye Mabelebele

**Affiliations:** 1Department of Agriculture and Animal Health, College of Agriculture and Environmental Sciences, University of South Africa, Florida 1710, South Africa; zahrabattal@gmail.com (Z.M.H.); manyelo.t.g@gmail.com (T.G.M.); letlhogonolo.selaledi@up.ac.za (L.S.); 2Department of Agricultural Economics and Animal Production, University of Limpopo, Sovenga 0727, South Africa; 3Department of Zoology and Entomology, Mammal Research Institute, Faculty of Natural and Agricultural Sciences, University of Pretoria, Hatfield 0028, South Africa

**Keywords:** antinutrients, feedstuffs, plant extracts, monogastric animals’ nutrition, tannins, health benefits

## Abstract

Over recent years, the monogastric animal industry has witnessed an increase in feed prices due to several factors, and this trend is likely to continue. The hike in feed prices is mostly due to extreme competition over commonly used conventional ingredients. For this trend to be subdued, alternative ingredients of both plant and animal origin need to be sourced. These types of ingredients are investigated with the aim of substituting all or some of the conventional compounds. However, alternative ingredients often have a double-edged sword effect, in that they can supply animals with the necessary nutrients although they contain antinutritional factors such as tannins. Tannins are complex secondary metabolites commonly present in the plant kingdom, known to bind with protein and make it unavailable; however, recently they have been proven to have the potential to replace conventional ingredients, in addition to their health benefits, particularly the control of zoonotic pathogens such as Salmonella. Thus, the purpose of this review is to (1) classify the types of tannins present in alternative feed ingredients, and (2) outline the effects and benefits of tannins in monogastric animals. Several processing methods have been reported to reduce tannins in diets for monogastric animals; furthermore, these need to be cost-effective. It can thus be concluded that the level of inclusion of tannins in diets will depend on the type of ingredient and the animal species.

## 1. Introduction

Monogastric animal production, in particular the poultry production sector, is growing continuously, driven mostly by the demand for meat and eggs. However, this rapidly growing industry and the increasing demand for poultry feeds have led to a considerable increase in feedstuff prices. The gap between demand and supply of balanced feed is expected to increase, and consequently increase the cost of production. On the other hand, the conventional feed ingredients such as maize, wheat and rice can no longer meet the poultry industry’s demand for feed. In addition, in-feed antibiotics have been used over a period of time as growth promoters, which positively aids in feed conversion rates and consequently reduces the cost. However, it was discovered recently that the inclusion of the antibiotics could leave residue in the meat and consequently cause resistance to some bacteria in humans [1]. These multifaceted challenges compelled the concerned researchers to look for alternative ingredients which can fill the gap. Tannins are considered valid alternatives to the conventional feed ingredients and as antipathogenic molecules, which can be used as an alternative ingredient.

Tannins are a group of polyphenolic compounds commonly found in the plant kingdom [2]. Because they are antimicrobial, antiparasitic, antiviral, antioxidant, and anti-inflammatory [2], they are considered valuable in that they can replace antibiotics in chicken feeds [2]. Although the use of tannins in monogastric animals’ feed has been discouraged over the years because of the antinutrient contents [3], recent studies have revealed that if tannins are used with caution, they can be of benefit to monogastric animals [4]. Tannins also can decrease the risk of livestock diseases and the spread of zoonotic pathogens. Current studies on the use of tannins in poultry production sector show favorable outcomes [5].

The mechanism with which tannins promote growth in the monogastric animals are not as clear as in ruminants [2]. The popular suggestion is that the inclusion of tannins in low concentrations leads to an increase in feed intake and consequently the performance of monogastric animals [2]. There is also a suggestion that the improvement in performance comes as a result of the creation of balance between the negative effects of tannins on feed palatability and nutrient digestion and the positive effects on promoting the health status of the intestinal ecology [2]. A study by [6] found that the condensed tannins available in the extract of grape seed reduces the fecal shedding of *E. Tenella*, and an increased growth performance of broiler chickens infected with *E. Tenella*.

To render tannins available to the monogastric animals, different processing methods to reduce the antinutrient effects are recommended. For example, the reduction of the tannin component of sorghum has improved its nutritional quality to become the closest alternative feed ingredient to maize in poultry diets [7]. Lately, different processing methods were introduced to reduce the tannin content in feed ingredients. The main methods used are cooking, dehulling, autoclaving, toasting, soaking, using wood ash, adding tallow, and using tannin-binding agents and enzymes. Hence, the aims of this review are(1) to elaborate on the use of tannins as alternative ingredient in monogastric animals’ feed; (2) to identify different structures and types of tannins; and (3) to identify successful processing methods to reduce the harmful effects of tannins.

## 2. Methodology

This review was conducted according to the reporting items for systematic reviews and meta-analyses (PRISMA) statement guidelines [8]. A comprehensive search was conducted to identify eligible studies. Databases, namely Web of Science, Science Direct, Google Scholar, PubMed and Wiley Online Database, were searched to obtain all relevant studies that were published before September of 2020. The search strategy used involved a combination of the keywords “tannins”, “alternative ingredients”, “monogastric animals”, “health benefits”, “condensed tannins”, “hydrolysable tannins”, “medicinal uses of tannins”, “antinutrients in tannins”, “antibiotic resistance” and “tannin processing methods”. Furthermore, the researchers narrowed their search to time scale 1977–2020 to include old and new studies to draw a comparison between the uses of tannins in monogastric animals with the current use. The search was not restricted by language, date, or study type. A total of 315 records were screened after removal of duplicates. Later, 218 records were excluded because they were irrelevant. The first draft articles were excluded for the following reasons: a) they did not cover the alternative feed subject, b) some of the articles did not adequately address the importance of tannins in livestock nutrition, c) some of the articles only focused on the undesirable antinutritional factors in the tannins. A total of 97 records were initially used to prepare the review.

In the second stage, extra records were searched to include ‘’antibiotic resistant strains’’ to add to the knowledge regarding antibiotic resistance and the health benefits of tannin. The overall number of records used to prepare this review was 122 records.

## 3. Structural Properties of Tannins

The physical and chemical properties of tannins differ according to the plant species [9]. Tannins are classified into two main parts—the hydrolysable tannins (HTs) and condensed tannins (CTs), also known as proanthocyanidins [10,11]. Hydrolysable tannins, as the name indicates, can be hydrolyzed by acids or enzymes. Their structure is characterized by a polyol core [12]. On the other hand, the condensed tannins are non-hydrolysable oligomeric and polymeric proanthocyanidins [13]. Condensed tannins are where the coupling of the single units is by positioning of *C*-4 of the first unit with *C*-8 or *C*-6 of the second unit [14]. The two most common condensed tannins are the procyanidins and the prodelphinidins [12]. There are three types of hydrolysable tannins, which include: gallotannins, ellagitannins, and complex tannins and condensed tannins, called procyanidins [15], (Figure 1). Gallic acid is mainly found in rhubarb and clove, while ellagic acid is found in eucalyptus leaves, myrobalans and pomegranate bark [16].

Further to this, recent research showed that tannins are produced inside an organelle named tannosome, which is believed to arise in cell plastids occurring in the green parts of plants that contain chlorophyll pigments. After creation, the tannosome is encapsulated in a membrane, and later transported to a plant vacuole for safe storage [17]. According to [12], the structures of the condensed tannins from different species can be differentiated based on the proportion of trihydroxylated subunits, ratio of *cis*/*trans* monomers, and the degree of polymerization. Figure 1 shows the classification of tannins into different classes.

## 4. Mode of Action and Functions of Tannins

Tannins are a complex group of polyphenolic compounds found in a wide range of plant species. They are characterized by astringency and tanning properties, which are believed to be associated with the higher molecular weight proanthocyanidins [20]. Hagerman [21] reported the molecular weight of tannins to be between 500 and 5000 Da. They are found in wood, bark, leaves and fruits; however, acacia species, which belong to the family of Leguminosae in the plant kingdom, are considered the most common sources of tannins [22]. Previously, harmful nutritional consequences have been attributed to tannins because they can precipitate proteins, inhibit digestive enzymes, and decrease the utilization of vitamins and minerals [23]. In addition, it was assumed that tannins are unabsorbable due to their high molecular weight and the ability to form insoluble structures with components of food such as proteins [24]. Hagerman et al. [11] reported that tannins in poultry feed affect dry matter intake and consequently the weight gain. Tannins that can be hydrolyzed are found in smaller amounts in plants, while the condensed tannins are found in abundance. The concentration of tannins is dependent on the plant genotype, tissue developmental stage, and the environmental conditions [12].

Biologically, tannins are significant in that they provide protection for the plant while still in the plant and have potential effects after the plant has been harvested [25]. In recent research, tannins have been proposed as an alternative to antibiotics because of the antimicrobial properties of tannins, which is the ability to inhibit extracellular microbial enzymes. In addition, hydrolysable tannins could be used in lieu of antibiotics, because bacteria such as *Clostridium perfringens* cannot develop resistance to them. However, their use in animal feed is discouraged because they impact nutrition negatively. Their use has been linked with lower feed intake and digestibility and leads to poorer animal performance.

Tannins have numerous applications that benefit humans. Some of the applications of tannins include their use as nutraceuticals to prevent, for example, cancer, cardiovascular disease, kidney disease, and diabetes [26]. They are also used for tanning leather, and manufacturing ink and wood adhesives. Medicinally, tannins are homeostatic, antidiarrheal, and a remedy for alkaloid and heavy-metals toxicity. In the lab, tannins are used as a reagent for protein detection, alkaloids, and heavy metals due to their precipitating properties. In the food industry, tannins are used to clarify wine, beer, and fruit juices. Other industrial uses of tannins include textile dyes, and as coagulants in rubber production.

## 5. Antibiotic Resistance in Animal Byproducts

Antibiotic resistance is a concern for animal welfare and as a hazard to public health since the contamination can be passed onto humans through the byproducts from animals. Although some contributing factors are unavoidable, such as the ability of bacteria to adopt to the changing environment [27], some of the factors are contributed by humans, such as the excessive use of antibiotics for growth promotion in farm animals [28]. For example, antibiotic resistant salmonella has been detected in meat [29]. Food animals are considered the main reservoir of antibiotic resistant bacteria, which can be transferred to humans through zoonoses and the food chain [30,31]. Some of the antibiotic resistant strains are presented in Table 1.

## 6. Medicinal Uses of Tannins

Tannins in plants are believed to function as chemical guards that protect the plants against pathogens and herbivores, as stated by [38]. Furthermore, the properties of tannins as antioxidants and reducing scavenging activities were also reported by [39]. The ability of tannins to chelate metals, their antioxidant activity, antibacterial action, and complexation are believed to be the mechanism of action behind tannins’ ability to treat and prevent certain conditions such as diarrhea and gastritis [40]. On the other hand, tannins’ mechanisms of antimicrobial activity include inhibition of extracellular microbial enzymes, deprivation of the substrates required for microbial growth, or direct action on microbial metabolism through inhibition of oxidative phosphorylation. The authors of [41] state that the antimicrobial properties of tannins are believed to be associated with the hydrolysis of ester linkage between gallic acid and polyols hydrolyzed after the ripening of many edible fruits, which enables the tannins to function as a natural defense mechanism against microbial infections. Table 2 demonstrates some of the medicinal uses of tannins [42].

## 7. Tannins as Adhesives

Tannins are used as a partial or complete substitute for phenols in wood adhesives in the form of tannin resin because of its phenolic structure [51]. The use of tannin adhesives was first successfully traded in South Africa in early 1970s [52]. It is documented that previous research in the field of fortified starch adhesives with wattle bark tannin was carried out in South Africa [53]. Mimosa tannin adhesives were used instead of synthetic phenolic adhesives to manufacture particleboard and plywood for external and marine applications [51]. In Kenya, the commercial wattle (*Acacia mearnsii*) is a well-known tannin-rich species and tannin-based adhesive [54]. Current industrialized technologies are based mostly on paraformaldehyde or hexamethylene tetraamine, which are considered more environmentally friendly [55]. The drive to create more environmentally friendly adhesives has led to different forms of research in the field; for example, the creation of corn-starch-tannin adhesives in a study by [56] in a bid to replace synthetic resins has shown that it has excellent structural stability.

## 8. Nutritive and Antinutritive Effects of Tannins

Tannins, commonly found in most cereal grains and legume seeds, as already indicated, are considered antinutritional factors that hamper the use of some feeds by monogastric animals. It has been reported that tannins bind protein, and as a result weakens protein digestion [57]. Tannins are blamed for the bitter taste of the feed, resulting in lowering feed consumption due to reduced palatability [58]. They are regarded as polyphenolic secondary metabolite; however, some reports have shown recently that low concentrations of some tannin sources can improve the nutrition and health status of monogastric animals [2]. Antinutrients are commonly known as natural or synthetic compounds that interfere with the absorption of nutrients. Condensed tannins are known to inhibit several digestive enzymes, including amylases, cellulases, pectinases, lipases, and proteases [59]. They have a major antinutritive effect that can influence the nutrient digestibility of lipids, starch, and amino acids negatively [60,61]. Tannins are a heterogeneous group of phenolic compounds, found in nature in many different families of plants. In Oakwood, Trillo, Myrobalaen and Divi-Divi they occur in almost every part of the plant, such as the leaves, fruits, seed, bark, wood and roots.

Supplementation of chestnut HT at the concentration of 0.5% and 1.0% on rabbit feed had no effect on growth performance [62]. However, [63] found different results when chestnut HT was included in rabbit feed at levels of 0.45% and 0.5%, as it increased feed intake and the live weight of rabbits. Similarly, [64] reported that adding 0.20% of chestnut, the tannin increased average daily gain and daily feed intake of broilers. The authors of [65] reported that when the sweet chestnut wood extract was used as a supplement at 0.07% and 0.02% for broiler chickens, no antinutritive activity was observed, and the crude ash, crude protein, calcium and phosphorus were not affected. The addition of tannic acid (HT) at a dietary level of 0.0125% and 0.1%, showed a negative impact on hematological indices and plasma iron of pigs [66]. According to [67], ideal digestibility of energy, protein, arginine and leucine were lowered in broiler chickens as dietary tannin levels rose to 20 g/kg diet and beyond, while phenylalanine and methionine were affected negatively only at tannin levels of 25 g/kg diet. In another study with broiler chickens [68], it was reported that the tannin content of 16 g/kg in red sorghum had no effect on phosphorus, calcium, and nitrogen retention in chickens. High-tannin sorghum treated with wood ash extract improves its nutritive value [69]. Tannins can act as a double-edged sword; therefore, a tannin content-specific solution could have an effect on their utilization. Although tanninferous feed and forages containing >5% tannin dry matter are not safe to be used as animal feed, low to moderate (<5% dry matter) is safe for animal consumption [59]. Table 3 shows the antinutritive and nutritive effects of tannins from different plant sources.

## 9. Influence of Tannins on the Productivity of Monogastric Animals

Tannins have been classified as an “antinutritional factor” for monogastric animals with negative effects on feed intake, nutrient digestibility, and production performance [1]. Currently, most researchers have revealed that some tannins can improve the intestinal microbial ecosystem, enhance gut health, and hence increase productive performance when applied appropriately in monogastric diets [62,70,75]. Strong protein affinity is a well-recognized property of plant tannins, which has successfully been applied to monogastric animals’ nutrition. However, adverse effects of high-tannin diets on monogastric animals’ performance have been reported by many researchers [71]. In monogastric animals, the main effects of tannins are related to their protein-binding capacity and reduction in protein, starch, and energy digestibility [76,77]. According to [10,78], dry matter intake, bodyweight, feed efficiency and nutrient digestibility were reduced when chickens were fed diets with tannins, whilst Ebrahim et al. [71] reported a decrease in body weight gain and feed intake. However, [72,75] reported no effects on growth performance and on egg weight, cell thickness or yolk color of layers. Several studies showed that low concentrations of tannins improved feed intake, health status, nutrition, and animal performance in monogastric farm animals [2,4,79].

According to [80], supplementing of pigs’ diet with 0.2% chestnut wood extract rich in tannins had no effect on growth rate, carcass traits or meat quality of pigs raised up to 26 weeks of age; whereas Bee et al. [81] reported that pigs that were fed diets rich in 3% of hydrolysable tannins from chestnuts showed no negative effects in terms of growing performance raised from day 105 until 165. The authors of [49] reported an increase in small intestinal villus height, villus perimeter and mucosal thickness in pigs that were fed diets having 3% of hydrolysable tannins from chestnuts. Moreover, [4] reported increased growth performance in pigs aged 23–127 days when fed chestnuts rich in tannins at the 0.91% supplementation level. According to [82], pigs have parotid gland hypertrophy and secrete proline-rich proteins in the saliva that bind and neutralize the toxic effects of tannins, which make them relatively resistant to tanniniferous diets without showing any negative effects as compared to other monogastric animals (Table 3).

In rabbits [62], no difference was observed in the performances of rabbits fed diets supplemented with up to 10 g of tannins from chestnuts. Moreover, they reported that no improvements were observed in health status, diet nutritive value, growth performance, carcass traits and oxidative stability of rabbits fed up to 400 g/100 kg of hydrolysable tannins originating from chestnuts. According to [83], rabbits fed diets with 4% of tanniniferous browsers of *Acacia karroo, Acacia nilotica* and *Acacia tortilis* showed no significant differences in intake and digestibility. Mancini et al. [84] also reported no significant difference in growth rate, feed intake or feed conversion ratio and carcass traits of rabbits fed a mixture of quebracho and chestnut tannins. Moreover, [85] observed no significant difference in growth rate, feed intake or feed conversion ratio of rabbits fed low-tannin sorghum grains. Thus, tannins, when included in monogastric animal diets, can have both positive and negative effects on animal performance, depending on the concentration. Therefore, it is important to minimize the inclusion or supplementation of feedstuffs containing high concentrations of tannins in monogastric animals, or to take measures to decrease their concentrations. In Table 4, the effect of tannins on productivity of monogastric animals is reported.

## 10. Processing Techniques Used to Reduce Effects of Tannins

Several processing techniques to reduce tannin levels in different feedstuffs, especially unconventional ingredients, have been suggested by most researchers [86,87]. Processing is an act of applying suitable techniques to reduce or eliminate tannins present in alternative feedstuffs. These techniques include enzyme supplementation, soaking, dehulling, alkali treatment, extrusion, and germination.

### 10.1. Enzyme Supplementation

Supplementation of enzymes to reduce the tannins content is an effective method, although it might not be the most economical. It is proven to reduce tannins better than other processing methods, such as soaking, dehulling, etc. Several studies have shown that enzyme supplementation has been effective in reducing tannins in alternative energy and protein feedstuffs [88,89]. A study by [88] found that treatment of sorghum with both polyphenoloxidase and phytase enzymes showed a decrease in hydrolysable and condensed tannins of 72.3% and 81.3% respectively. Moreover, [89] reported a decrease in both hydrolysable and condensed tannins by 40.6%, 38.92% and 58.00% respectively when sorghum grains were treated with the three enzymes tannase, phytase and paecilomyces variotii.

### 10.2. Soaking

Soaking is one of the cheapest traditional methods which animal nutritionists have used for many years. A study found that the addition of sodium bicarbonate, prolonged time of soaking, or higher temperature have proved to be effective during the soaking process [90]. Kyarisiima et al. [69] reported that high-tannin sorghum soaked in wood ash extract showed a decreased level of tannins without lowering the nutrient content of sorghum grains. Authors stated that tannin level did not only decrease with the soaking technique, but also with roasting. The decrease in tannins during soaking may result from leaching into the soaking water [77]. Moreover, [91] reported a decrease of about 73–82% in velvet beans.

### 10.3. Dehulling

Dehulling is a process of reming the outer coat/hull of a seed [92]. Most seeds of alternative feedstuffs have seed coats/hulls which are normally concentrated with tannins. If tannins are removed, feedstuffs have shown to have a significant increase in protein digestibility and protein content in legume seed meal. The authors of [93] reported that dehulling reduced tannins in chickpea without lowering protein digestibility, whereas in faba beans a 92% decrease of tannins occurred with dehulling [94].

### 10.4. Extrusion

The extrusion method is used to decrease levels of tannins in feedstuffs. According to [95], extrusion cooking is a high-temperature, quick process in which starchy food materials are plasticized and cooked by a combination of moisture, pressure, temperature, and mechanical shear. Extrusion has shown the ability to inactivate antinutritional elements [96,97,98]. For example, [99] reported that extrusion showed a significant reduction in tannins with minimum oil loss in flaxseed meal. The authors of [100] reported that lentil splits showed a reduction in tannins after treatment by using extrusion techniques. Moreover, [101] reported reduction to the extent of 34.52% to 57.41% in sorghum.

### 10.5. Germination

During the germination process, complex sugars are converted into simple sugars [91]. Tannin content has been shown to be reduced by the germination process, which is one of the cheapest methods. A maximum reduction in tannins of up to 75% has been observed when pearl millets were treated by using the germination method [102]. Rusydi and Azlan [103] observed a reduction of 57.12% when peanuts were treated by using germination. The reduction of tannins may improve the nutritional quality of feedstuffs. Thus, processing techniques may help to remove or reduce tannin levels in different feedstuffs, which might be favorable for animal production (Table 5).

### 10.6. Cooking

Cooking is considered important in reducing antinutrients activities in tannins. As stated by [104], cooking reduces the antinutrients present in tuber crops like cocoyam.

### 10.7. Auticlaving

Autoclaving is found to be one of the most effective methods in the elimination of antinutrients, although it might not be cost effective because of its reliance on electricity [105].

### 10.8. Grinding

Grinding is considered an effective method in reducing the tannin content because it increases the surface area which in turn reduces the contact between tannins and the phenolic oxidase in the plant [106,107].

## 11. Health Benefits of Tannins in Monogastric Animal Production

Tannins are plant extracts that can be used as additives in monogastric animal feed to control diseases [1]. In vitro studies have shown that most tannins have antiviral, antibacterial and antitumor properties [15]. Tannins have shown a favorable outcome in the preferment of gut health when used with other antimicrobials as growth-promoting factors (AGP) such as probiotics [1]. Condensed tannins extracted from green tea or quebracho have shown to have some antimicrobial substances [108]. However, [109] reported that condensed tannins may have less effect than hydrolysable tannins in controlling *Campylobacter jejuni* in the presence of high concentration of amino acids. Moreover, tannins derived from chestnuts (*Castanea sativa*) can inhibit the in vitro growth of *Salmonella typhimurium* [110]. Several in vitro studies have revealed that polyphenols of the procyanidins (CT) have an antioxidant property while tannic acid has anti-enzymatic, antibacterial and astringent properties, as well as constringing action on mucous tissues [111]. The ingestion of tannic acid causes constipation, so it can be used to treat diarrhea in the absence of inflammation [112]. Kumar et al. [69] reported that the tannin content of 16 g/kg in red sorghum had no effect on certain animal welfare parameters of broiler chickens. Similarly, globulin, protein, plasma albumin, phosphorus, glucose, calcium, and uric acid levels were not affected, even when maize is replaced 100% with red sorghum. However, mild histopathological changes in kidney and liver tissues, as well as high cell-mediated immune response, were detected when raw red sorghum containing 23 g tannins/kg was fed to the same group of broiler chickens. The supplementation of purple loosestrife (*Lythrum salicaria*) in rabbits has led to a significant increase in the total white blood cells and higher concentrations of volatile fatty acids and acetic acid, therefore a low level of loosestrife supplementation (<0.4%) has been suggested to gain health benefits and prevent adverse effects on animal health and performance [113].

Farmatan tannin concentrations of 0.05%, 0.025% and 0.0125% can inhibit the growth of *Clostridium perfringens* by more than 54-fold [114]. Another in vitro study was conducted by [108] to evaluate the effects of tannins from chestnuts and quebracho, or a combination of both, on *Clostridium perfringens.* All three products reduced the presence of *C. perfringens*. When the comparative analysis was conducted, it was discovered that the concentrations of quebracho tannin were more effective in inhibiting the growth of *C. perfringens* as compared to chestnut tannin. Commensal bacteria such as *Bifidobacterium breve* or *Lactobacillus salivarius* are very useful and their growth or presence should not be inhibited by the tannin. Kamijo et al. [115] reported that ellagitannins isolated from *Rosa rugose* petals have some antibacterial activities against pathogenic bacteria such as *Salmonella* sp, *Bacillus cereus*, *S. aureus* and *E. coli* but they had no effect on beneficial bacteria. Most in vitro results are supported by in vivo experiments that the inclusion of tannin in monogastric animals can lower the occurrence and severity of diarrhea [116]. However, the efficiency of adding tannins that shows robustness in inhibiting pathogens in in vitro studies needs to be evaluated further in the experimental set-up (in vivo) involving poultry and pigs. These disparities in terms of types of tannins that are efficient in combating certain pathogens warrant further research. Table 6 shows different health benefits of tannins in monogastric animals.

## 12. Conclusions

In the quest to find alternative feed ingredients in the production of monogastric animals, the effects of tannins have proven to be of value. Tannins can be beneficial in both as feed ingredients and a valuable ingredient in animal health. Although tannins contain antinutrients, different processing methods have proved to be effective in the reduction or elimination of these antinutrients. This review has provided extensive literature on the benefits and impacts of tannins in poultry production. Furthermore, it has elaborated on the different processing methods which can be employed to reduce the negative effects of tannins. The methods chosen should be cost-effective, easy to use and should not defeat the purpose of alternative feed ingredients. Even though tannins can act as feed additives, their inclusion level will depend on the source, age and species of poultry. Thus, future research should focus on the optimum tannin inclusion level in poultry and more cost-effective processing methods, especially for small-scale poultry keepers who mostly utilize these alternative feed ingredients. The development of more convenient readily available products of tannins ready to be incorporated in the monogastric animal feed is encouraged.

## Figures and Tables

**Figure 1 molecules-25-04680-f001:**
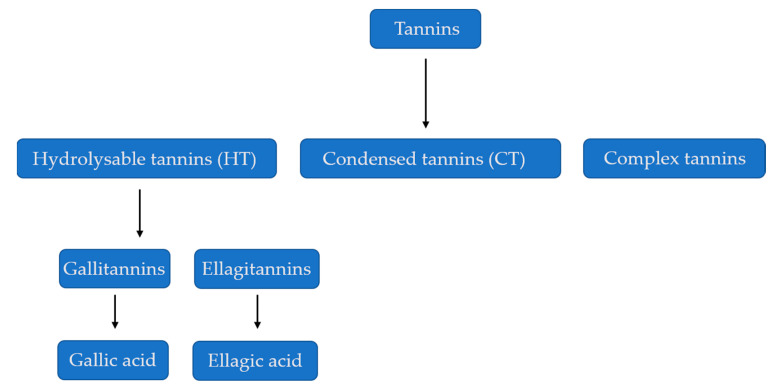
Classification of Tannins. Sources: [18,19].

**Table 1 molecules-25-04680-t001:** Examples of antibiotic resistant strains in animal by-products.

Antibiotic Resistant Strains	Animal Product	References
*Staphylococcus*	Cattle meat and milk	[32]
*Salmonella*	Poultry meat	[33]
*Campylobacter*	Poultry meat	[34]
*Escherichia coli*	Cattle Liver and minced turkey meat	[35]
*Escherichia coli*	Poultry meat	[36]
*Escherichia coli*	Poultry meat	[37]

**Table 2 molecules-25-04680-t002:** Uses of tannins as medicinal sources and industrial agents.

Components	Medicinal Uses	References
Sweet chestnut extracts	*Escherichia coli*, *Bacillus subtilis*, *Salmonella enterica* serovar Enteritidis	[43]
Extract of chestnut shell	Enteritidis, *Clostridium perfringens*, *Staphylococcus aureus*, and *Campylobacter jejuni*	[44]
Gall nuts	Treatment of diarrhea and dermatitis	[45]
*Acacia Nilotica*	Antimutagenic and cytotoxic effects	[46]
Sweet chestnut extracts	Reduction of Salmonella infection	[47]
Quebracho Tannins	Reduction of worm eggs counts and inhibition of development of nematodes and lungworms	[48]
Chestnut extracts	Control of *Clostridium perfringens*	[49]
Pine needles and dry oak leaves	Control of coccidian infection	[50]

**Table 3 molecules-25-04680-t003:** Nutritive and antinutritive effects of tannins in monogastric animals.

Plant Source/Tannin	Animal (Monogastric)	Concentration/Application	Effects	References
Chestnut (*Castanea*) HT	Swine/pig	1%, 2% and 3%	Liver not affected. Changes in the intestine: villus height increased, mucosal thickness and villus perimeter; reduced large intestinal apoptosis and mitosis	[70]
Sweet chestnut wood extract	Chickens (broilers)	0.07% and 0.2%	No antinutritive effects	[65]
Tannic acid (TA)	Chickens (broilers)	1% Tannic acid different climatic conditions	Better quality of fatty acid profile of breast muscle of broilers	[71]
Chestnut (*Castanea*) HT	Chickens (layers)	0.20%	Increased monounsaturated fatty acid and reduced cholesterol content of eggs	[72]
Chestnut tannin extract (*Castanea sativa* Miller) HT	Chickens (layers)	2 g/kg	Unsaturated fatty acids increased; cholesterol significantly decreased: −17% in WLT and −9% in MUT	[73]
High-tannin red sorghum (*Sorghum vulgaris*) HTS	Chickens (broilers)	16 g/kg (reconstituted red sorghum)	Utilisations of phosphorus, nitrogen and calcium retention were similar	[68]
Chestnut (*Castanea*)	Pigs	0%, 5%, 10% and 15%	Reduction in digestibility of dry matter, crude protein, ether extract, crude ash and tannin decreased linearly (*p* < 0.05) with increasing chestnut meal supplementation	[74]

**Table 4 molecules-25-04680-t004:** Effects of tannins on productivity of monogastric animals.

Tannin Concentrations	Tannin Source	Monogastric Animal	Influenced/Affected Parameter	References
0.16–0.19%	Chestnut	Pigs	Increased growth performance	[4]
0.71–1.5%	Chestnut	Pigs	No effect on feed intake, body weight gain and carcass traits; reduced feed efficiency	[81]
1–3%	Chestnut	Pigs	Increased small intestinal villus height, villus perimeter and mucosal thickness	[70]
5–10%	Grape pomace	Broilers	No effect on growth performance; increased oxidative stability and polyunsaturated fatty acids content of thigh meat	[75]
1%	Tannic acid	Broilers	Decreased body weight gain and feed intake; improved the fatty acid profile of breast muscle	[71]
	Chestnut	layers	No effect on egg weights, cell thickness or yolk colour; reduced cholesterol content	[72]
0.45% and 0.5%	Chestnut	Rabbits	Increased live weight gain and feed intake of rabbits	[79,86]
0.5% and 1.0%	Quebracho and chestnut	Rabbits	Had no effect on growth performance	[62,84]
4%	*Acacia karroo*, *Acacia nilotica* and *Acacia tortilis*	Rabbits	No significant differences in intake and digestibility	[83]

**Table 5 molecules-25-04680-t005:** Different processing techniques used to reduce the effects of tannins in alternative feedstuffs.

Processing Technique	Feedstuff	Effectiveness	References
Enzyme supplementation	Sorghum	The enzyme tannase reduced both hydrolysable and condensed tannins by 40.6%	[89,90]
Dehulling	Chickpeas	Reducing tannin level without lowering the nutrient content of the grain	[94]
	Faba beans	Reduced about 92% of tannins	[95]
Soaking	Sorghum	Reducing tannin level without lowering the nutrient content of the grain	[69,78]
	Velvet beans	Decreased about 73–82% of tannins	[92]
Alkali treatment	Sorghum	Reducing tannin level without lowering the nutrient content of the grain	[78]
Extrusion	Flaxseed	Significant reduction of tannins with minimum oil loss in flaxseed meal	[99]
	Lentils	Reduced the tannin content in lentil splits	[100]
	Sorghum	Reduction to the extent of 34.52% to 57.41%	[101]
Germination	Pearl millets	Maximum reductions in tannins up to 75%	[102]
	Peanuts	Reduction of tannins by 57.12%	[103]
Cooking	Cocoyam	Reduction of antinutrients in tuber crops	[104]
Autoclaving	Sorghum	Reduction to the extent of 34.52% to 57.41%	[101]
Germination	Pearl millets	Maximum reductions in tannins up to 75%	[102]
	Peanuts	Reduction of tannins by 57.12%	[103]

**Table 6 molecules-25-04680-t006:** Health benefits of tannins in monogastric animals.

Plant Source/Tannin	Animal/Monogastric	Application Rates	Health Benefits	References
Chestnut tannin (HT)	Chickens	0, 250, 500 and 1000 mg/kg	250 mg/kg reduced number of *E. coli* and coliform bacteria in small intestine. Greatest number of *Lactobacillus* observed in supplementation of 1000 mg/kg	[49]
Purple loosestrife (*Lythrum salicatia*)	Rabbit	0.2%, 0.4% and 0.3%	Increased total white blood cells in rabbit	[113]
Chestnut (HT)	Chickens (broiler)	0.15% to 1.2%	Reduced bacteria in the gut. *Clostridium perfringens* (*Eimeria maxima*, *Eimeria tenella* and *Eimeria acervulina*)	[117]
Grape pomace (CT)	Pigs	2.80%	Reduction in the absorption of mycotoxins in the gastrointestinal surface	[118]
Grape pomace (CT)	Chickens (broiler)	6%	Increased commensal bacteria (*Lactobacillus*) and decreased the counts of clostridium bacteria in ileal content	[118]

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
