# Peer review of "The Effects of Tannins in Monogastric Animals with Special Reference to Alternative Feed Ingredients"

_molecules, 2020, doi:10.3390/molecules25204680_

Round 1
Reviewer 1 Report
The authors present a review of tannin effects in monogastric animal feed. The review is informative but the relevance and the correlation between the tannin feeding and the search for alternative feeding for monogastric animal industry is somewhat unclear. The authors seem to suggest that the animal industry searches for tannin content alternative feed yet, also, suggest that there are already several reports of anti-nutritional factors for this component. The introduction needs revision to better present the intention and most of all the relevance of the review. The article needs revision mostly on the section 3 and 4 of the discussion. It is not suggested that the authors use images or tables not owned by them, even in reviews, except in cases of extreme need. The main source of information used in reviews must come from the filtered articles main findings. Chemical structures can be drawn by the authors.
L 17: “replacing all or some conventional ones” this sentence is vague
The abstract does not display the main findings of the review, it goes straight into the conclusion. The abstract needs revision
Keywords should be different from the words present in the title
L 30 – 36: Reference is lacking
Methodology lacks the number of articles retrieved and filter used to narrow the articles first retrieval
L 79 – 102: The relevance and continuity of this section needs improvement. The figures are not relevant to the review and are not owned by the authors.
L 106 – 107: Units are missing
L 109 – 110: “The name ‘tannin’ originated from the Celtic word, which means oak, a traditional source for tannin-containing extracts for preparation of leather” There is no relevance for this information in a section named “Mode of action and functions of tannins”
L 136 - 137: Reported by whom?
Table 1 is not owned by the authors
Reviewer 2 Report
The aims of the paper is elaboration of the tannins use, identify different structures and types of 68 tannins, and the identification of successful processing methods propose to reduce the harmful effects of tannins.
- Please put more details about novel studies related to the antibiotic residue in the meat and the risk related to that.
- Please put references to the statement "tannins are not that all good news as they may cause harmful effects on animal performance, particularly in monogastric animals, causing damage on gut villi".
- Please improve the quality of Fig 1
- In my opinion in Fig 1 should be some examples of the tannins groups
- It is visible that structures in Fig 2 are copied. Authors showed appropriate agreement for copy? It has very low quality
- Page 5 "proteins [23]. [11] reported". Informations are missing. Sentance may not has beginning [11]. Correct in the whole text
- Please define the antimicrobial properties of tannins (which strain)
- "(1.6 X 107 spores/mL)." is it correct? (page 9)
- Page 9 "a-galactoside" what did authors mean?
- Conclusions does not include the summarization. Please also improve the further studies and perspectives.
- In my opinion the title should be - a minireview. For review paper is to short and should include at least 120 references
Round 2
Reviewer 1 Report
The mansucript has improved and some corrections were made.
A specific line in the methodology section should be revised still.
L 83-84: "records were excluded because they were irrelevant" This section must be revised prior to acceptance, explain in greater detail what is this "irrelevance" and list the correct reasons for filtering
one example would be : "The articles of the first draft were exclude by the following reasons: Pertained to topics other than animal feeding; did not address feeding aspects of tannins; did not address monogastic animals"
The above is just an examples, but it is expected a list of the filters, just as showed above
Reviewer 2 Report
Thank you for comments. In my opinion paper is ready to be published.
Author Response
Thank you very much for the positive feedback.